# The Impact of the COVID-19 Pandemic on Ophthalmic Outpatient Care in a Tertiary Care Center in Riyadh

**DOI:** 10.3390/healthcare10091654

**Published:** 2022-08-30

**Authors:** Majed S. Alkharashi, Heba M. Alsharif, Faisal A. Altahan, Ahmad W. Alrashed, Moath Abdulghani

**Affiliations:** 1Department of Ophthalmology, College of Medicine, King Saud University, Riyadh 11461, Saudi Arabia; 2College of Medicine, King Saud University, Riyadh 11461, Saudi Arabia

**Keywords:** COVID-19, pandemic, ophthalmology, outpatient

## Abstract

In this paper, we measured the impact of a full COVID-19 lockdown on ophthalmic patients after a period of lockdown in Saudi Arabia, from March to September 2020. A cross-sectional analytical study was carried out on 180 patients who had their appointments delayed or canceled due to the lockdown. Data was collected from electronic medical records and patients via voice calls using a validated questionnaire that were analyzed using a multivariable binary regression analysis. The results show no statistically significant mean difference in visual acuity when comparing pre- and post-lockdown measurements. The median number of appointment cancellations/delays per patient was two, and the estimated delay for the first canceled appointments was equal to 178.8 days. Of the cohort studied, 15.4% of patients faced delays in necessary surgical and therapeutic interventions; 22.1% of patients sought eye care at other institutions due to the delay, and 15% of those were seen by doctors unspecialized in ophthalmology. The odds of dissatisfaction with care were higher in patients who experienced cancellations in a surgical procedure and patients who experienced difficulty in obtaining medications. In conclusion, the pandemic hampered ophthalmic patients’ access to medications. Subjective visual outcomes of patients were also negatively affected; however, the change in objective visual parameters was not statistically significant.

## 1. Introduction

The recent Coronavirus-19 (COVID-19) pandemic was a global catastrophe that forced countries around the world to undergo lockdown [1], which heavily impacted healthcare systems around the world [2,3]. Healthcare utilization in general declined by about a third during the pandemic [4]. Many elective surgical interventions were delayed, and non-urgent medical care across all specialties was postponed [5]. As outpatient clinics are not usually a place of urgent care, they were particularly affected by the pandemic [6,7]. One of the major roles of outpatient care which was halted during this period was routine screening and preventative services, aimed at early detection prevention of disease. Childhood vaccinations took a big hit as a result of this, with orders from Centers for Disease Control and Prevention dropping by nearly 11 million doses in 2020 from previous yearly averages [5]. Reports of an increase in late-stage cancer diagnosis and cancer mortality were also attributed to the delay of healthcare during the pandemic [8,9].

Ophthalmology has one of the busiest outpatient clinics among medical specialties [10], and with many appointments canceled, many patients with chronic ophthalmological conditions requiring regular follow-ups no longer received access to the routine services they needed [11]. This may have an impact on the course of their disease, or it may lead to neglect of their existing conditions, delay detection of potentially sight-threatening complications or even delay vision-saving interventions, all of which may play a role in worsening of visual outcomes [12].

With the advancement and accessibility of technology comes the opportunity to utilize it in the medical context in the form of telemedicine. Although it is not a sufficient alternative to in-person clinic visits to diagnose ocular conditions, telemedicine does however serve as a modality for provisional screening and management of ocular conditions without the risk of contracting the infection [12].

A full national lockdown took place in Saudi Arabia during the pandemic, which forced clinics around the country to limit the number of patients in outpatient clinics in compliance with social distancing [13,14,15]. This obliged King Abdulaziz University Hospital, which is a government-funded free healthcare provider, to delay around 400 appointments scheduled during this period.

The aim of this study is to investigate the impact of the COVID-19 lockdown on outpatient ophthalmic care, and to evaluate the consequences of delayed healthcare delivery on overall eye health. Furthermore, this article aims to shed light on patient satisfaction with care during the pandemic.

## 2. Materials and Methods

This is a quantitative, observational retrospective analytical study. This study was carried out in Riyadh city, Saudi Arabia. Data were obtained from patient records as well as through a questionnaire administered through a voice call with the patients who had their appointments canceled or delayed during the national lockdown in Saudi Arabia during the peak of the COVID-19 pandemic from June to September 2020. All patients in the ophthalmic outpatient department in King Abdulaziz University Hospital (KAUH) whose appointments were canceled or delayed during the lockdown period were included.

A pilot study was conducted on 18 individuals to assess the validity of a questionnaire that was tailored to the authors’ objectives and to calculate the response rate, and the questions were modified accordingly. A validated survey was divided into three parts: the first part was basic demographic information about the patient; the second had to do with information about the patient’s ophthalmic diagnosis, their documented visual acuity pre and post the lockdown, their follow-up clinic and the medications currently prescribed by their ophthalmologist if any, all of which were obtained from the electronic medical record. The third part had to do with patients’ subjective ophthalmic outcomes, their health-seeking behavior during the lockdown, and whether the patient resorted to external health providers and private hospitals to meet their needs in drug refills, eye checkups or even surgery because of the delay that occurred. Patients were also asked about their utilization of telemedicine through phone applications. LogMar values were calculated from Snellen fractions by using an Excel conversion tool that was developed by Moussa et al. [16]. Lastly, an open-ended question was directed to the patient to observe the overall subjective feedback of the patient’s experience with the care provided during the pandemic. Informed consent was orally obtained from every participant before administering the survey and patient confidentiality was respected as no names or identification were included. Participants did not receive any incentives or rewards.

## 3. Results

One hundred and eighty patient records were included in the study. A retrospective review of the patients’ medical records was conducted coupled with a telephone call with each patient to collect the data regarding their experience with the outpatient clinics during and after the pandemic. Descriptive data analysis results were obtained for the patients’ sociodemographic characteristics, and most of the patients (53%) were females. Analysis of patient age distribution showed that 10.5% of them were <20 years of age, whereas almost half of the patients (48.6%) were elderly (aged ≥ 60 years). The rest were between 20 and 60 years of age. Most of the patients resided in Riyadh, the capital city of Saudi Arabia, and 27.6% of them resided in urban areas around the capital city or in other provinces.

The patients’ ophthalmological medical history and appointment characteristics are shown in Table 1. Most patients had a diagnosis of glaucoma (34.6%), and the second most common reason for follow-up was cataract (16.8%). The patients were followed up by six different eye clinics: 33.1% of them were followed up by the cornea clinic, 37.6% of them by the glaucoma clinic, 3.3% by the laser clinic, 5% of them by the pediatric clinic and the remainder 17.7% by the retina clinic. The findings also showed that the majority of patients (85.1%) were scheduled for follow-up visits, 8.3% were new patients and only 1.7% of them were referrals from other hospitals/clinics.

Analysis of the mean visual acuity of both eyes by LogMar values before and after the lockdown revealed the following: according to the WHO classification of visual acuity, it was initially found that 68% of the patients had mild to no visual impairment, another 13.8% of them were considered to have moderate visual impairment and 3.9% had severe impairment. A further 14.4% of them were considered to have very severe visual impairment.

A paired samples t-test showed no statistically significant mean differences between patients’ overall mean LogMar scores (of both eyes) when pre- and post-lockdown values were compared (*t* = 1.75, df =180, *p* = 0.080), although the patients’ visual acuity scores appeared to be slightly better after the pandemic.

Table 2 shows that the mean number of appointment cancellations the patients had experienced was equal to 1.55 times, with a median cancellation time of 2. The estimated delay time (in days) for the patients’ first canceled appointments was equal to 178.8 days, with a median delay of 105.4 days. In addition, the data analysis findings show that 4.4% of the patients had experienced a postponement of necessary interventions (such as intravitreal injections) due to the COVID-19 lockdown; moreover, 11% of them had experienced a rescheduling of necessary surgeries that were already booked until another later date. Regarding the prescribed medication and eye drops: the majority of patients (64.9%) were prescribed eye lubricants, 49.1% of patients were on glaucoma eye drops, and 7.9% of them were taking topical ophthalmic steroid drops. Only 3.5% were on prescribed oral steroids and immunomodulatory medications for uveitis treatment, and 7% of them were on a course of ophthalmic antiviral and antibiotic drops. The patients were asked to answer No or Yes as to whether they had experienced difficulties obtaining their necessary medication refills and 23.2% of them indicated they had encountered difficulties, but 53% of those who faced difficulties stated they were able to obtain their medications despite the difficulties and obstacles encountered with the lockdown. For those who had obtained their medications, 47% of them had received their required medications from their hospital pharmacy and another 48% of them had resorted to buying them from external sources/pharmacies.

The patients were asked to indicate how their treating clinic communicated their changes in appointment dates; 98.2% of them advised that their clinics had sent them a text message, 28.8% were contacted via phone call by the hospital staff and 0.6% were sent an email reminder or notification. However, 3.7% had to physically come to the hospital to communicate with their caregivers or to pick up their prescriptions. The findings from the analysis showed that 1.7% of the patients used an online clinical application or telemedicine to communicate with their caring clinic. It was also shown that 22.1% of patients went to another private/governmental hospital to get treatments or eye checkups, but when asked to indicate the type of external clinical services they needed, some of the patients (3.9%) would not disclose that information. A small fraction (5%) of them went to other local governmental hospitals, and 13.3% went to private clinics

The patients who sought care at external hospitals were asked to state the type of management they had received. The findings showed that 21.2% had received surgical interventions, one third of them (33.3%) received top-up medications and most of them (66.7%) underwent routine eye checkups. When asked about their current chief complaints, patients reported the following: 13.2% had blurred vision, 21.1% had eye dryness, 34.2% complained of eye pain and 28.9% reported eye redness. One tenth (10.4%) of patients experienced eye itchiness and 5.3% had an ocular burning sensation. The patients were asked to subjectively describe their eye condition after the pandemic, and most of them (74%) experienced no changes or noted some improvement; however, 26% of them reported a worsening of their visual condition. In terms of patient satisfaction with care during the pandemic, 28.2% of the patients were dissatisfied with the appointment delays they experienced, although the majority were generally satisfied.

To better understand what may explain the patients’ final subjective experience of worsened eye condition during the lockdown, a multivariate binary logistic regression analysis was used to assess the patients’ odds of feeling that their eye condition had worsened following appointment cancellation with their sociodemographic and other clinical outcomes and attributes. The findings from the multivariate analysis, shown in Table 3, suggested that female patients were found to be 2.84 times more likely to feel their eye condition had worsened compared with their male counterparts, *p* = 0.008. Furthermore, the patients’ age in years correlated significantly with their odds of reporting worsening in their eye condition during the delay period; for each one-year rise in the patient’s age, their odds of experiencing a worsened eye condition following appointment cancellation rises by a factor equal to 1.021 times higher on average, *p* = 0.031. In addition, the findings from the analysis model show a significant prediction that on average, and by considering the other predictor variables in the analysis model accounted for, the patients who were prescribed lubricant eye drops were 0.254 times less likely to experience eye condition worsening than patients who were not on lubricant drops, *p* = 0.002. However, patients who sought routine eye checkups at external clinical services did not correlate significantly with their odds of worsened eye condition, but the patients who received treatment or underwent an ophthalmic procedure in external clinics during the lockdown were found to be 6.28 times more likely to experience worsened eye condition post the lockdown on average compared with people who had not received external clinical eye care, *p* = 0.001. The patients’ baseline right (OD) and left (OS) visual acuity levels did not converge significantly on their odds of experiencing worsened eye condition at later stages post lockdown.

The patients’ overall satisfaction with their clinical appointments and cancellations was also regressed against their sociodemographic and clinical outcomes (Table 4), to better understand what may explain their dissatisfaction with their outpatient department clinical appointments. The findings showed that the patients’ sex, age and type of caring clinic, as well as the reason for their clinical appointments, did not correlate significantly with their odds of dissatisfaction with their clinic. However, the patients who experienced delays or cancellations in a surgical procedure were found to have 3.050 times greater odds for dissatisfaction with their clinical appointment compared with those who had no surgical procedural cancellations, *p* = 0.049. Moreover, the patients who experienced difficulty in obtaining their top-up medications during the lockdown period were found to be 4.21 times more likely to be dissatisfied with their clinical experience compared with the patients who had no medication refilling difficulties or delays, *p* = 0.001. In addition, the patients who had a subjectively worsened eye condition following appointment cancellation were found to be 2.937 times higher on average compared with patients who had reported stable or improved eye conditions, *p* = 0.008.

The multivariate binary logistic regression analysis was used to assess what may explain the patients resorting to external clinics and services in light of delayed appointments noting that all of these patients had undergone a delay in their appointments. The findings from the multivariate analysis model, shown in Table 5, indicated that the patients’ age and sex did not converge significantly on their odds of going to external services due to their primary appointment cancellations. Furthermore, the patients who had experienced cancellation of surgical procedures by their clinics were found to be slightly more likely to attend other external medical services; however, the correlation between experiencing canceled surgical procedures with their odds of using other external services was not statistically significant, *p* = 0.066. Difficulty in obtaining refill medications did not correlate with the odds of seeking external medical services, but there was found to be a significant prediction that the patients who had subjectively reported worsened eye conditions were 4.10 times more likely to use external services compared with patients with stable or improved eye conditions, *p* = 0.004, accounting for the other predictor variables.

Additionally, the resulting multivariate analysis findings showed that the patients who experienced the mean number of appointment cancellations correlated significantly and positively with their odds of using external medical services: as the patients’ number of canceled appointments rises by one cancellation, on average, their odds of resorting to external clinical services tends to rise by a factor equal to 1.79 times higher on average, *p* = 0.012. Figure 1 clearly demonstrates that as the patients’ number of canceled appointments (on the *x*-axis) rises, their adjusted probability of referring to external clinics tends to rise incrementally on the *y*-axis accordingly. In addition, the analysis model demonstrated that the patients who faced difficulty obtaining their lubricant drop top-ups were found to be 4.62 times more likely to use external medical services compared with those who did not face difficulties obtaining their lubricant drop top-ups, *p* = 0.002.

## 4. Discussion

The study showed that the COVID-19 pandemic lockdown had a significant impact on the ophthalmic patients at KAUH outpatient clinics. This is clear from the prolonged delay that most patients faced with their appointments, which was estimated to be 178.82 days. Notably, most patients (53%) faced difficulties with acquiring their medications during this time. This was distressing to these patients, who were shown to be 4.21 times more likely to report subjective worsening of their condition having encountered difficulties with medication refills. Contrary to what the authors hypothesized, the visual acuity pre and post the lockdown did not show significant differences in the analysis, nor were patients who experienced a documented decline in their visual acuity more likely to be dissatisfied with care.

It is worth mentioning that KAUH is a governmental institution that provides free care to all its patients. This might explain why patients who had to resort to external pharmacies to get their necessary medications were more likely to be dissatisfied with care. Concerningly, of those who faced difficulties with acquiring medications, only 53% managed to attain them, but the remaining 47% were not able to. Such deferral of treatments can have major effects on some conditions, such as glaucoma, uveitis and eye infections. We recommend that clear instructions are given to patients on how to get their required medications during times of limited access to hospitals, such as during this pandemic, and we propose that every hospital should have a hotline dedicated to this cause.

After a thorough review of the literature, it was clear to the authors that no study has addressed the difficulties ophthalmic patients faced during the lockdown period (delays, medications, worsening symptoms) while also assessing how patients dealt with such hardships. No current studies in the middle-eastern peninsula have analyzed the ophthalmic patients’ response to appointment delays during the COVID-19 pandemic lockdown. However, one study in the UK looked into the psychosocial effects of the lockdown on ophthalmic patients [17], and the results demonstrated that 45.9% of a total 325 respondents expressed fear of further sight loss due to the delay in treatment, and in addition, 39.2% mentioned that their eye disease had become more difficult to cope with during the lockdown. This is not far from the 26% of people who reported a worsening of their eye condition in the current study. This is expected, as a delay of 173 days, combined with a delay of necessary interventions and treatments, would be more than sufficient to aggravate the visual and psychological condition of patients.

Most researchers, on the other hand, were concerned with reporting the reduction in ophthalmic patient flow during the pandemic, as with a study done in an ophthalmic center in India [18] and another study in a medical retina center in Italy [19], all of which reported a significant drop in patient flow during the pandemic. Similar findings were noted in Massachusetts, USA [20,21].

Interestingly, the use of telemedicine was much lower than expected in the current study (1.7%). This in contrast with the 1800% increase noted in a UK study, which highlighted the receptivity of ophthalmic patients to telemedicine during the COVID-19 pandemic [22]. An explanation perhaps could be that clinics at KAUH have yet to establish a telemedicine portal that is easily accessible to both patients and doctors. Another explanation could be that most patients from the study at hand were older adults (48.6%) and the elderly are not well equipped regarding the use of technology. In addition, 27.6% of the patients in this study also resided in rural areas, which further supports this explanation.

## 5. Conclusions

The lockdown due to COVID-19 had a major impact on ophthalmic outpatient care as hundreds of appointments were cancelled. In addition, patient satisfaction, medication accessibility and subjective visual outcomes of patients were negatively affected; however, the objective visual outcomes did not show a statistically significant change. The study showed that most patients who sought external ophthalmic care sought it from physicians untrained in ophthalmology, which is alarming. Therefore, we recommend that hospitals provide clear instructions and guidelines to patients and increase utilization of telemedicine to improve the quality of healthcare services in case of future lockdowns and pandemics.

## 6. Study Limitations

The inclusion of both subjective and objective visual outcomes and addressing the effect of the pandemic from the patients’ perspective is one of the strengths of this study. To the best of our knowledge, no similar studies have been carried out in Saudi Arabia, highlighting an unmet need in the literature which this study aims to fulfill. An additional strength on this study is the fact that it was carried out in one of the biggest specialized ophthalmology centers in Saudi Arabia, and thus we believe this study provides a suitable representation of the state of ophthalmic care in Saudi Arabia during the pandemic. Nevertheless, one of the limitations of this study is the fact that patients were surveyed a few months after their appointment cancellation. This introduces recall bias, although an effort was made to reduce such biases by comparing between subjective and objective data. Another limitation is that these results represent the effects of the lockdown on ophthalmic patients in one center only, and cannot be generalized to all ophthalmic clinics in Riyadh. Lastly, the subjectivity of the visual outcomes the patients reported during the pandemic might have been exaggerated or even understated by some. Nonetheless, subjective visual complaints are not to be disregarded as they are central in ophthalmic patient care.

## Figures and Tables

**Figure 1 healthcare-10-01654-f001:**
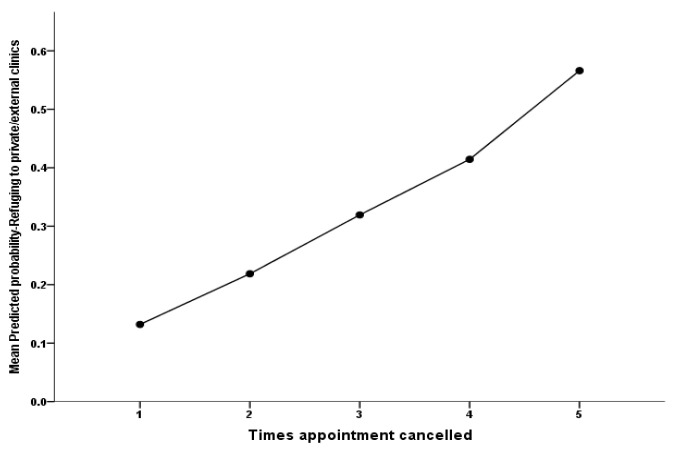
The association between the patients’ number of postponed appointments and their model of predicted mean probability of referring to external services.

**Table 1 healthcare-10-01654-t001:** Descriptive analysis of the patients’ ophthalmologic medical history findings.

	Frequency	Percentage
Ophthalmologic diagnosis		
Keratitis	8	4.5
Glaucoma	62	34.6
Conjunctivitis	3	1.7
Cataract	30	16.8
Diabetic retinopathy (NPDR * + Macular edema)	22	12.3
Cataract surgery (status post phacoemulsification)	19	10.6
Status post keratoplasty	5	2.8
Refractive errors	6	3.4
Uveitis	12	6.7
Other diagnoses	16	8.9
Caring outpatient eye clinic		
Cornea clinic	60	33.1
Glaucoma clinic	68	37.6
Laser clinic	6	3.3
Pediatric clinic	9	5
Retina clinic	32	17.7
Uveitis clinic	6	3.3
Reason for the clinical appointment		
Follow-up	154	85.1
New patient	15	8.3
Referral	9	5
Scheduled procedure	3	1.7
Patients’ overall LogMar level before pandemic (OU)		
Mild or no visual impairment	114	63
Moderate visual impairment	30	16.6
Severe visual impairment	18	9.9
Very severe visual impairment	19	10.5
Patients’ overall LogMar level after pandemic (OU)		
Mild or no visual impairment	123	68
Moderate visual impairment	25	13.8
Severe visual impairment	7	3.9
Very severe visual impairment	26	14.4

* Non-proliferative diabetic retinopathy.

**Table 2 healthcare-10-01654-t002:** Descriptive analysis of the patients’ ophthalmologic clinical appointment and cancellation outcomes.

	Mean	Standard Deviation
Frequency of canceled appointments	1.55	0.833
Delayed time (duration) in days	178.82	105.43
	Frequency	Percentage
Canceled/postponed scheduled interventions		
No	173	95.6
Yes	8	4.4
Canceled/postponed scheduled surgeries		
No	161	89
Yes	20	11
Prescribed eye medications		
Glaucoma drops	56	49.1
Eye lubricant drops	74	64.9
Eye steroid drops	9	7.9
Uveitis medications	4	3.5
Antibiotics/Antivirals	8	7
Did the patient encounter any difficulties obtaining their refill medications during the pandemic?		
No	139	76.8
Yes	42	23.2
Despite the difficulties, were the patients able to obtain their refill medications?		
No	85	47
Yes	96	53
If the patient obtained their medications, from where did they obtain them?		
Hospital pharmacy	66	67.3
External pharmacy	48	49
Method of communication by clinic (telemedicine—voice calls—SMS *) to intervene with delays		
The clinic sent the patient an SMS	160	98.2
Personally attended to inquire about appointment status	6	3.7
Telephoned the patient	42	25.8
Sent an email	1	0.6
Does the patient use an online clinical application?		
No	178	98.3
Yes	3	1.7
Did the patient seek another clinic/service outside the hospital due to the canceled service?		
No	141	77.9
Yes	40	22.1
What kind of external clinical service did the patient visit?		
No external clinic visits	141	77.9
Did not disclose	7	3.9
Governmental hospital/emergency department	9	5
Private clinic	24	13.3
Was the attending physician in the external clinical service a trained ophthalmologist?		
No	147	81.2
Yes	34	18.8
Interventions administered by external clinical services		
Surgical interventions	7	21.2
Refilled the patient’s required medications	11	33.3
Underwent eye checkups	22	66.7
Prescribed refractive correction	4	12.1
Current patients’ subjective outcomes (chief complaint)		
Blurred vision	5	13.2
Eye dryness	8	21.1
Eye pain	13	34.2
Tearing	11	28.9
Eye redness	11	28.9
Itchiness in the eyes	4	10.4
Eye burning sensation	2	5.3
Other eye complaints	3	7.9
Post-lockdown subjective visual outcome		
Stable/improved	134	74
Worsened	47	26
Overall patients’ satisfaction with the clinical services during the pandemic		
Satisfied	130	71.8
Dissatisfied	51	28.2

* Short messaging service.

**Table 3 healthcare-10-01654-t003:** Multivariate logistic binary regression analysis of the patients’ odds of experiencing worsened eye condition during pandemic time.

	Multivariate Adjusted Odds Ratio	95% C.I. * for OR ^†^		*p*-Value
		Lower	Upper	
Female sex	2.838	1.311	6.145	0.008
Age (years)	1.021	1.002	1.041	0.031
Prescribed lubricant drops	0.254	0.108	0.596	0.002
Underwent eye checkup at an external clinic	0.318	0.078	1.290	0.109
Visited/received treatment at an external clinic	6.218	2.103	18.382	0.001
Before COVID-19 (OU) eye LogMar	0.866	0.489	1.536	0.623
Before COVID-19 (OS) eye LogMar	1.025	0.627	1.675	0.922
Constant	0.077			<0.001

Note: Dependent outcome variable = Patients’ dissatisfaction with the service post COVID-19 cancellations of appointments (No/Yes). * Confidence Interval ^†^ Odds Ratio.

**Table 4 healthcare-10-01654-t004:** Multivariate logistic binary regression analysis of the patients’ odds of dissatisfaction with clinical service during pandemic time.

	Multivariate Adjusted Odds Ratio	95% C.I. * for OR ^†^		*p*-Value
		Lower	Upper	
Female sex	1.006	0.480	2.107	0.988
Age (years)	1.004	0.988	1.021	0.611
Type of follow-up clinic	1.233	0.978	1.556	0.077
Reason for appointment	1.298	0.741	2.272	0.362
Experienced surgical cancellation	3.050	1.004	9.268	0.049
Experienced difficulty in obtaining medications from the clinic during lockdown.	4.209	1.842	9.615	0.001
Experienced worsened eye condition post-lockdown	2.937	1.324	6.514	0.008
Received medical treatment at an external clinic	2.106	0.853	5.198	0.106
Constant	0.048			<0.001

Note: dependent outcome variable = patients’ dissatisfaction with the service post COVID-19 cancellations of appointments (No/Yes). * Confidence Interval ^†^ Odds Ratio.

**Table 5 healthcare-10-01654-t005:** Multivariate logistic binary regression analysis of the patients’ odds of resorting to other (external) clinical services due to appointment cancellations associated with the pandemic lockdown.

	Multivariate Adjusted Odds Ratio	95% C.I. * for OR ^†^		*p*-Value
		Lower	Upper	
Female sex	0.509	0.214	1.212	0.127
Age (years)	0.986	0.967	1.005	0.139
Had an eye surgical procedure canceled by the clinic due to COVID-19 lockdown	3.095	0.929	10.308	0.066
Experienced difficulty refilling medications	1.340	0.524	3.427	0.541
Experienced worsened eye condition during the pandemic	4.104	1.585	10.629	0.004
Number of appointments canceled	1.795	1.138	2.830	0.012
Need to refill eye lubricant drops	4.608	1.755	12.102	0.002
Constant	0.062			<0.001

Note: dependent outcome variable = patients needed to go to external clinics for eye treatment (No/Yes). * Confidence Interval ^†^ Odds Ratio

## Data Availability

The data is available on the EHR system at KAUH.

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
