# Peer review of "The Impact of the COVID-19 Pandemic on Ophthalmic Outpatient Care in a Tertiary Care Center in Riyadh"

_healthcare, 2022, doi:10.3390/healthcare10091654_

Round 1

Reviewer 1 Report

The Authors presented an article assessing the impact of the COVID-19 pandemic on ophthalmic care in Saudi Arabia. In conclusion, the authors emphasize negative impact on ophthalmic patients. 

Major:

In 11 “after a period of complete lockdown in Saudi Arabia” please specify dates

In 22 you conclude “the pandemic significantly affected ophthalmic patients”. How do you get to this conclusion if objective visual outcome did not show statistically significant change? P-value?

In 45 “delay around 400 appointments scheduled during this period” and in 75 “180 patients were included”. To how many people did you administer the questionnaire? What was the response rate? When did you perform it? (302, recall bias) How many patients does the KAUH outpatient clinics schedule every day? Please correctly specify the methods. Very small size. Several biases

143-151 this paragraph is very messy, punctuation marks and parentheses are missing

All the results are duplicated in the tables and the text, please review all this entire section avoiding repetitions.

292 “major impact”?

297 “Clear hospital instructions and guidelines on seeking medical care is essential and the increased utilization of telemedicine might improve the quality of healthcare services” How do you support this conclusion with your arguments?

I miss some newer references, almost all citations are from 2020.

Reviewer 2 Report

Thank you for the opportunity to revise this interesting manuscript on the impact of the COVID-19 pandemic on ophthalmic outpatient care.

I believe the topic is very important, and the manuscript well-presented.

In particular, I found the study design appropriate, and the methods clearly described. The findings are well-presented and support the discussion and conclusions.

Overall, I really enjoyed reading the manuscript, and I have only a minor suggestion the authors could consider.  The only amendment is related to the introduction section, which does not draw a complete international panorama.

In fact, the introduction section could be improved by adding some more references in relation to the COVID-19 impact on the healthcare systems. Accordingly, the authors mentioned what happened with regard to cardiovascular and rheumatic patients (see references 2 and 3), but I believe this is not sufficient to introduce the heavy impact of the COVID-19 pandemic. For example, authors should consider the impact on oncologic patients and the consequences related to the low vaccination rate, with particular regard to the childhood vaccination programs (see, for example, I): https://doi.org/10.1016/S2468-2667(22)00111-6; II) https://doi.org/10.3390/ijerph18115642).

I hope the authors will follow the suggestions as I believe the manuscript will be well-received by the readers of the Journal.

Round 2

Reviewer 1 Report

I appreciate the effort put into this revision. My concerns have been addressed.

However, I highly recommend the authors to include in the manuscript methods section the clarification you have given in the authors' response that I couldn't find in the revised manuscript regarding response rate. (e.g. "the appointments that were delayed during the lockdown period were 368 ... the questionnaire was administered to 180 of them ... Our response rate is 49% which is quite acceptable given the difficult circumstances of the pandemic....")

All the best.
